# Hybrid Active Learning with Uncertainty-Weighted Embeddings

**Yinan He**[*][†]                                                    *yinan.he.688@gmail.com*
*Nanyang Technological University, Singapore*

**Lile Cai**[*]                                                      *caill@i2r.a-star.edu.sg*
*Institute for Infocomm Research (I²R), A\*STAR, Singapore*

**Jingyi Liao**                                               *liao_jingyi@i2r.a-star.edu.sg*
*Institute for Infocomm Research (I²R), A\*STAR, Singapore*

**Chuan-Sheng Foo**                              *foo_chuan_sheng@i2r.a-star.edu.sg*
*Institute for Infocomm Research (I²R), A\*STAR, Singapore*

*Reviewed on OpenReview:* *https://openreview.net/forum?id=jD761b5OaE*

## Abstract

We introduce a hybrid active learning method that simultaneously considers uncertainty and diversity for sample selection. Our method consists of two key steps: computing a novel uncertainty-weighted embedding, then applying distance-based sampling for sample selection. Our proposed uncertainty-weighted embedding is computed by weighting a sample's feature representation by an uncertainty measure. We show how this embedding generalizes the gradient embedding of BADGE so it can be used with arbitrary loss functions and be computed more efficiently, especially for dense prediction tasks and network architectures with large numbers of parameters in the final layer. We extensively evaluate the proposed hybrid active learning method on image classification, semantic segmentation and object detection tasks, and demonstrate that it achieves state-of-the-art performance.

## 1 Introduction

There has been a recent resurgence of interest in using active learning (AL) to address the annotation burden for training deep neural networks. By selecting only an informative subset of samples to label for model training, AL methods can significantly reduce the cost of data annotation while maintaining satisfactory performance (Ren et al., 2021). Depending on the criterion used to select samples, AL methods can be broadly categorized into uncertainty-based, diversity-based and hybrid methods. Uncertainty-based methods (Wang et al., 2016; Gal et al., 2017; Beluch et al., 2018; Yoo & Kweon, 2019; Liu et al., 2021) select samples that the model is most uncertain about, while diversity-based methods (Sener & Savarese, 2017; Sinha et al., 2019; Caramalau et al., 2021) select a subset of samples that are representative of the training data distribution. Employing uncertainty-based sampling alone can result in redundancy in selected samples, as similar samples are likely to have similarly high uncertainty values. On the other hand, diversity-based sampling may waste the annotation budget on distinctive yet easy samples. Hybrid approaches aim to overcome these drawbacks by considering both uncertainty and diversity for selecting samples.

Various methods have been proposed for hybrid AL. The first approach to combine uncertainty and diversity sampling is to apply them sequentially, *e.g.*, apply uncertainty sampling to select the top-k most uncertain samples (where k is larger than the budget size) and then enforce diversity sampling to select a subset that

---

[*]Joint first authors
[†]Work done during internship with I²R

satisfies the budget (Yang et al., 2017). A second approach is to sum up uncertainty and diversity terms in the acquisition function (Yang et al., 2015; Liu & Ferrari, 2017; Paul et al., 2017; Cai et al., 2021b). A third approach is to perform diversity-based sampling on uncertainty-aware features (Ash et al., 2019; Kim et al., 2021). In particular, the BADGE (Ash et al., 2019) method has been shown to achieve state-of-the-art performance on image classification tasks (Ji et al., 2023). BADGE applies diversity sampling on gradient embeddings – the first-order derivative of the loss function with respect to the parameters of the last layer. Gradient embeddings of uncertain samples have a large magnitude; this correlation between uncertainty and embedding magnitude allows BADGE to select a batch of both uncertain and diverse samples.

While BADGE performs well on image classification, three challenges arise when applying BADGE to other types of tasks. First, the computation of gradient embedding can be expensive for dense prediction tasks like segmentation and for network architectures that have a large number of parameters in the last layer. Second, the gradient embedding is only derived for cross-entropy loss in the BADGE paper (Ash et al., 2019); the form of gradient embedding with other loss functions can be complicated and hard to interpret. Finally, computing gradient embeddings on unlabelled data requires pseudo-labels as proxies for true labels. While predicted labels have been shown to be effective proxies for classification tasks, this is not the case for regression tasks – directly using predicted values as proxies for the ground truth results in embeddings with zero magnitudes (details in Section 3.1).

In this work, we propose a simple yet effective embedding method to address these challenges. Our method is motivated by the observation that the gradient embedding of a neural network model can be decomposed into two terms. The first term can be interpreted as an uncertainty measure of the current model on the sample, while the second term is the feature representation extracted from the penultimate layer of the network. We propose to keep this general two-term structure while relaxing the constraint of computing the embedding as the derivative of the loss. We achieve this by instead using an embedding that weights the feature representation by an arbitrary uncertainty term. The proposed uncertainty-weighted embedding enables diversity sampling in an uncertainty-aware feature space like gradient embedding, without requiring a specific form of loss function. As a result, our proposed embedding can be used with any learning task while also being much faster to compute than BADGE embeddings. We integrate our proposed uncertainty-weighted embedding within a distance-based sampling framework to perform hybrid AL, and demonstrate the effectiveness of the proposed AL method on a wide range of tasks and datasets.

Our contributions can be summarized below:

- We propose a novel uncertainty-weighted embedding for hybrid AL. We show that the proposed uncertainty-weighted embedding can be viewed as a generalized version of gradient embedding that does not rely on the derivative of the loss function.

- We integrate the uncertainty-weighted embedding within a distance-based sampling framework to perform hybrid AL. We extensively evaluate this hybrid AL method on three tasks: image classification, semantic segmentation and object detection, and demonstrate that it obtains state-of-the-art results on all three tasks.

## 2 Related Work

### 2.1 Deep Active Learning

As a promising technique to alleviate the data annotation burden in deep learning, AL has recently attracted significant attention from the research community. Depending on the criterion used to select unlabelled data, AL methods can be categorized into uncertainty-based, diversity-based and hybrid methods.

**Uncertainty-based methods** Methods falling into this group select samples based on uncertainty measure. Some commonly used uncertainty measures include entropy (Shannon, 1948), BvSB (Best versus Second Best) (Joshi et al., 2009) and LC (Least Confidence) (Wang et al., 2016). BALD (Gal et al., 2017) selects samples that maximize the mutual information (*i.e.*, reduction in uncertainty) between predictions

and model posterior. Instead of pre-defining the uncertainty measure, Yoo & Kweon (2019) propose to employ a loss prediction module to learn the uncertainty. Beluch et al. (2018) show that deep ensembles lead to more calibrated uncertainties and better AL performance. ISAL (Liu et al., 2021) estimates a sample's influence on model parameters via influence function and selects samples that can provide the most positive influence.

**Diversity-based methods**   Such methods aim to select a subset of samples that are representative of the training data distribution. CoreSet (Sener & Savarese, 2017) employs K-Center-Greedy algorithm to select a core set such that the largest distance between a data point and its nearest neighbor in the core set is minimized. VAAL (Sinha et al., 2019) learns a latent space in an adversarial manner and selects samples that are most different from labelled ones. Caramalau et al. (2021) build a graph convolutional network on the features extracted from the target model to learn a better feature space for active selection.

**Hybrid methods**   The various methods of combining uncertainty and diversity sampling in AL can be grouped into three categories. The first is to apply the two sampling strategies sequentially. Yang et al. (2017) first select top-k samples with the largest uncertainty scores and then perform diversity sampling on the k samples to remove redundancy. Random sampling can be applied before uncertainty sampling to improve the diversity of the selected samples (Beluch et al., 2018; Yoo & Kweon, 2019). The second approach is to combine uncertainty and diversity terms by summation in acquisition function. USDM (Yang et al., 2015) formulates the batch mode AL as a quadratic programming problem, with the unary term optimized for uncertainty and the pairwise term accounted for diversity. Paul et al. (2017) build a graph to represent the unlabelled pool and employ a submodular objective function to select samples that minimize the joint entropy of the nodes of the graph. The third approach is to perform diversity-based sampling on uncertainty-aware features. BADGE (Ash et al., 2019) represents each sample by its gradient embedding and applies K-Means++ seeding algorithm to select samples with large magnitude. TA-VAAL (Kim et al., 2021) conditions the discriminator of VAAL on the learned loss to obtain an uncertainty-aware diversity score for sample selection. Our method falls into the third category of hybrid AL. Compared to BADGE, our method does not rely on a specific form of loss function, and has significantly reduced computation costs.

## 2.2   Active Learning for Visual Tasks

**AL for image classification**   AL for image classification has been extensively studied in the literature and most methods can be classified into the aforementioned three categories, namely uncertainty-based methods (Gal et al., 2017; Beluch et al., 2018; Yoo & Kweon, 2019; Liu et al., 2021), diversity-based methods (Sener & Savarese, 2017; Sinha et al., 2019; Caramalau et al., 2021) and hybrid methods (Ash et al., 2019). Ji et al. (2023) conducts a comprehensive benchmarking on 7 state-of-the-art AL methods for image classification, and shows that BADGE wins the overall ranking. We compare with BADGE in our experiments.

**AL for semantic segmentation**   Semantic segmentation aims to assign a class label to each pixel in an image. Based on the granularity of data selection for annotation, previous methods can be classified into image-based methods (Yang et al., 2017; Sinha et al., 2019) and region-based methods (Mackowiak et al., 2018; Casanova et al., 2020; Cai et al., 2021b). Yang et al. (2017) aggregate the pixel-level uncertainties and features to obtain an image-level score and feature vector and select images to label greedily. VAAL (Sinha et al., 2019) learns a model to score each image based on its difference from labelled data. Region-based methods divide an image into regularly-shaped (Mackowiak et al., 2018; Casanova et al., 2020; Cai et al., 2021b) or irregularly-shaped (Cai et al., 2021a) regions for selection. We demonstrate our method with both image-level and region-level selection.

**AL for object detection**   Object detection network consists of a backbone feature extractor, a class prediction branch and a location prediction branch. Depending on whether the method relies on some specific architecture design, methods can be divided into architecture-specific (Roy et al., 2018; Yuan et al., 2021) and architecture-agnostic (Yoo & Kweon, 2019; Agarwal et al., 2020; Wu et al., 2022) approaches. Roy et al. (2018) measures the uncertainty of a prediction by the disagreement of convolution layers that predict for the same object in one-stage object detector. MI-AOD (Yuan et al., 2021) construct a detector

with two class prediction heads and one multiple instance learning head, and measure uncertainty by the discrepancy of the two classifiers. The learning loss-based approach LLAL (Yoo & Kweon, 2019) has been employed for object detection by learning the total loss of the detection model. CDAL (Agarwal et al., 2020) exploits the contextual information captured in the predicted probability distribution to enforce contextual diversity in selected samples. EnmsDivproto (Wu et al., 2022) considers bounding box level similarity to generate image-level uncertainty scores and to reject redundant samples. CALD (Yu et al., 2022) measures the uncertainty of a sample by its prediction consistency between original and augmented views. Our method is architecture-agnostic and can be readily applied to both one-stage and two-stage object detectors.

## 2.3 Uncertainty Weighting for Active Learning

The idea of using uncertainty to weight feature representation has been explored for active domain adaptation (Prabhu et al., 2021) and semi-supervised active learning (Buchert et al., 2022). Prabhu et al. (2021) proposed to perform K-Means clustering on the feature space, with the center of each cluster being updated by uncertainty-weighted averaging. Buchert et al. (2022) proposed a consistency-based embedding space for active learning where the feature of each image is weighted by the prediction consistency of the image under various augmentations. Unlike these works which are only applicable to image classification tasks, our proposed hybrid AL framework is general and can be readily applied to a wide range of tasks.

# 3 Method

We consider pool-based batch-mode active learning, where a batch of samples is selected from a pool of unlabelled samples in each cycle. Our method, described in Algorithm 1, starts with a randomly selected batch and proceeds iteratively until the annotation budget is met. Each iteration involves two main components: embedding extraction and sample selection. We first provide a recap on BADGE with discussions on its strengths and weaknesses. We then describe each component of our method in detail, followed by illustrations on how it can be applied to three different visual tasks, namely, image classification, semantic segmentation and object detection.

---

**Algorithm 1** Active Learning with Uncertainty-Weighted Embeddings

---

**Require:** Budgets $\{ b_1,\ b_2,\ \ldots,\ b_T \}$; Dataset $\mathcal{D}$
1: Randomly select $b_1$ samples to form the initial labelled pool $\mathcal{L}$ and train the initial model $h_1$
2: Initialize the unlabeled set $\mathcal{U} = \mathcal{D} \setminus \mathcal{L}$
3: **for** $t = 2,\ \ldots, T$ **do**
4:     $\mathcal{B}_t = \emptyset$
5:     For all samples, compute uncertainty-weighted embeddings based on Eq.(4) using model $h_{t-1}$
6:     For each sample $i \in \mathcal{U}$, compute its minimum distance to $\mathcal{L}$: $D(i) = \min_{j \in \mathcal{L}} ||uwe(i) - uwe(j)||$
7:     **while** $|\mathcal{B}_t| < b_t$ **do**
8:         $s = \arg\max_{i \in \mathcal{U}} D(i)$
9:         $\mathcal{B}_t \leftarrow s$
10:         For each sample $i \in \mathcal{U}$, update its minimum distance to $\mathcal{L} \cup \mathcal{B}_t$ as: $D(i) = \min(D(i), ||uwe(i) - uwe(s)||)$
11:     **end while**
12:     Annotate the samples in $\mathcal{B}_t$
13:     $\mathcal{L} \leftarrow \mathcal{L} \cup \mathcal{B}_t$
14:     $\mathcal{U} \leftarrow \mathcal{U} \setminus \mathcal{B}_t$
15:     Train model $h_t$ on $\mathcal{L}$
16: **end for**
17: **return** model $h_T$

---

## 3.1 Recap of BADGE

BADGE selects a batch of diverse and uncertain samples by applying K-Means++ seeding algorithm on gradient embeddings. The gradient embedding of each sample is computed as the first-order derivative of the loss function with respect to parameters in the final layer. As ground truth label is unknown for unlabelled data, BADGE uses the model's current prediction as a proxy of the true label. Let $x$ denote a sample and $h$ denote a L-layer neural network parameterized by $(\theta_1, \ldots, \theta_L)$. With softmax activation and

cross-entropy loss, the gradient embedding is computed as:

$$ge(x) = ((p_i - \mathbb{1}(\hat{y} = i)) \cdot f(x; \theta_{1:L-1}))_{i=1}^K, \tag{1}$$

where $p_i$ is the predicted probability for class $i$, $\hat{y} = \arg\max_{i \in [K]} p_i$ is the pseudo label for $x$, $f(x; \theta_{1:L-1})$ is the output of the network's penultimate layer, and $K$ is the number of classes. The gradient embedding is essentially scaling the feature presentation $f(x; \theta_{1:L-1})$ by the predicted probabilities. When the model is certain for a sample, $p_{\hat{y}}$ would be relatively large and $||ge(x)||$ would be small. The correlation between uncertainty and gradient embedding magnitude allows BADGE to select a batch of both uncertain and diverse samples by K-Means++ seeding algorithm.

BADGE has been shown to perform well on image classification task. However, when applying BADGE to other types of tasks, there are three challenges. First, the computation of gradient embedding can be expensive for dense prediction tasks like segmentation, as the derivative needs to be evaluated on every pixel; it is also expensive for network architectures that have a large number of parameters in the last layer, *e.g.*, for RetinaNet (more details in Section 5.1). Second, it is common to employ loss function that is customized for data distribution in some tasks, *e.g.*, focal loss for object detection and dice loss for medical image segmentation. The form of gradient embedding under such loss functions is complicated and hard to interpret. Third, for regression task the pseudo label can not be obtained by simply discretizing the prediction as classification task does. Let $\hat{y}$ denote the predicted value and $y$ be the ground truth. It can be derived that the gradient embedding of $L_2$ loss takes the form $2(\hat{y} - y)\partial\hat{y}/\partial\theta_L$. Directly using $\hat{y}$ as a proxy for $y$ results in embedding of zero magnitude for all samples.

## 3.2 Uncertainty-Weighted Embedding (UWE)

We note that the gradient embedding of a neural network can be decomposed into two terms using the chain rule:

$$ge(x) = \frac{\partial \ell}{\partial z} \frac{\partial z}{\partial \theta_L}, \tag{2}$$

where $z = \theta_L \cdot f(x; \theta_{1:L-1})$ is the last layer output before applying activation function. The second term $\frac{\partial z}{\partial \theta_L}$ is essentially the feature representation extracted from the penultimate layer and is independent of loss function:

$$\frac{\partial z}{\partial \theta_L} = \frac{\partial \theta_L \cdot f(x; \theta_{1:L-1})}{\partial \theta_L} = f(x; \theta_{1:L-1}). \tag{3}$$

The first term $\frac{\partial \ell}{\partial z}$ depends on the form of loss function and activation function. For the two cases discussed above, $\frac{\partial \ell}{\partial z}$ can be derived as below:

1. Cross-entropy loss with softmax activation:
   $\frac{\partial \ell}{\partial z} = (p_i - \mathbb{1}(y = i))_{i=1}^K$;

2. $L_2$ loss with identity activation:
   $\frac{\partial \ell}{\partial z} = 2(\hat{y} - y)$,

where $y$ and $\hat{y}$ is the ground truth and prediction, respectively. We observe that in both cases the magnitude of $\frac{\partial \ell}{\partial z}$ can be interpreted as an uncertainty measure of current model on the sample. In the first case, the magnitude of $(p_i - \mathbb{1}(y = i))_{i=1}^K =$ would be large if model is uncertain on the prediction (as $p_y$ is small); in the second case, if we approximate true value $y$ by the average of an ensemble of models, the magnitude of $(\hat{y} - y)$ is a measure of dispersion and would be large when uncertainty is high. This motivates us to relax the constraint of computing the embedding as the derivative of loss function, and to generate an embedding by directly weighting the feature by uncertainty. Specifically, we define a function $u(x)$ to measure the uncertainty of current model on sample $x$, and a feature extractor $f(x)$ by taking the intermediate output of the network. With the uncertainty measure and feature representation, the proposed uncertainty-weighted embedding is defined as:

$$uwe(x) = u(x) \cdot f(x). \tag{4}$$

The benefits of the proposed embedding are two-fold. First, it shares the same property of gradient embedding that the magnitude is positively correlated to sample's uncertainty. As illustrated in Fig. 1, weighting the feature by uncertainty imposes positive correlation between the resulting UWE's magnitude and uncertainty. This property facilitates diversity sampling in an uncertainty-aware feature space by distance-based selection (detailed in Section 3.3). Second, it does not rely on a specific form of loss function. Both the uncertainty measure $u(x)$ and feature extractor $f(x)$ can take any off-the-shelf method instead of being tied to a specific form or layer. This makes our method flexible and versatile for different types of tasks.

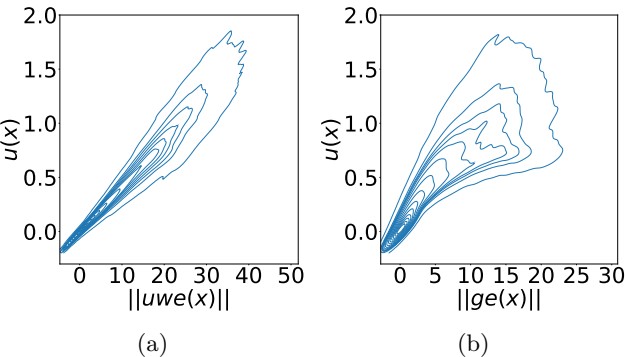

Figure 1: Weighting the feature by uncertainty imposes positive correlation between the magnitude of UWE and uncertainty. (a) Correlation between the magnitude of UWE and uncertainty (Pearson correlation coefficient $\rho = 0.98$). (b) Correlation between the magnitude of gradient embedding and uncertainty ($\rho = 0.90$). Density plots are drawn on CIFAR-10 using the model trained in the first batch for uncertainty (measured by entropy) and feature extraction.

### 3.3 Distance-based Sampling

With uncertainty-weighted embeddings, selecting samples of large magnitude is more likely to obtain samples of large uncertainty. However, directly selecting the top-k largest ones can result in redundancy as similar samples will have similarly high values. We thus propose to employ distance-based sampling to perform selection. Distance-based sampling selects samples based on its distance to previously selected ones. As illustrated in Figs. 2a and 2b, samples of larger magnitude also have larger Euclidean distance between them, which are more likely to be selected; while similar samples or samples of small magnitude have small Euclidean distance between them, which are unlikely to be repeatedly selected. Therefore, by applying distance-based sampling on uncertainty-aware feature space, we are able to obtain a set of both uncertain and diverse samples.

There are two popular distance-based sampling methods, K-Center-Greedy algorithm (KCG) used in CoreSet (Sener & Savarese, 2017), and K-Means++ seeding algorithm (KM++) (Arthur & Vassilvitskii, 2006) used in BADGE (Ash et al., 2019). Both methods decide the next sample to select based on its distance to its nearest neighbor in previously selected ones. The only difference is that KCG deterministically selects the sample with the largest distance in each iteration, while KM++ selects samples probabilistically with probability proportional to the squared distance to the selected samples. The probabilistic nature of KM++ makes it prone to select samples from high density region, which corresponds to region of small magnitude in the UWE space (as illustrated in Fig. 1, samples of low uncertainty have small magnitude and thus concentrate around the origin (high density), while samples of high uncertainty have large magnitude and would spread out in space far from the origin (low density)). Figure 2c visualizes the samples selected by KCG and KM++. KCG is able to select samples of larger uncertainty and diversity. We thus adopt KCG as the selection method (see Sec. A.1 for more discussion on selection methods).

### 3.4 Application to Visual Tasks

**Image classification** Deep models for multi-class image classification employ softmax activation function on logits and output a probability distribution vector. Sample uncertainty can be measured by entropy. Let $x$ denote a sample, $p(y = i|x)$ the predicted probability for class $i$, and $K$ the number of classes, entropy is

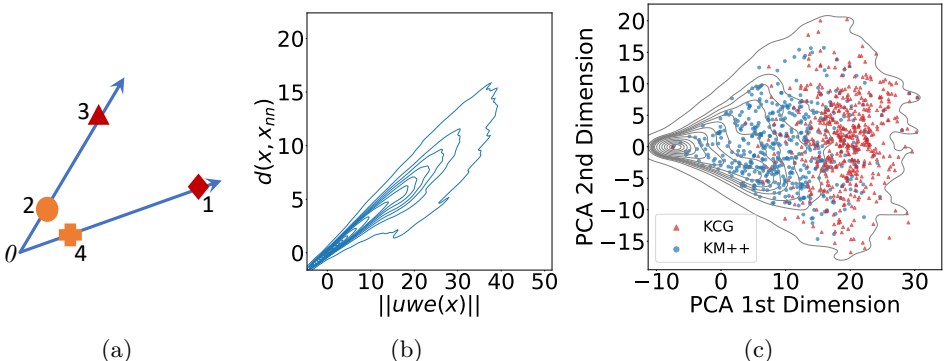

Figure 2: Distance-based sampling on uncertainty-weighted embeddings. (a) Illustration of the selection order for 4 samples by KCG. Samples of small magnitude (in orange) have smaller distance between them and will not be repeatedly selected (for budget $\leq 3$). (b) Correlation between $||uwe(x)||$ and $d(x, x_{nn})$ ($\rho = 0.88$). $d(x, x_{nn})$ represents the distance of sample $x$ to its nearest neighbor $x_{nn}$ in the dataset. (c) Visualization of the samples selected by KCG and KM++ on the density plot of the UWE space. Plots are drawn on CIFAR-10 using the model trained in the first batch for uncertainty (measured by entropy) and feature extraction, and the visualized samples in (c) are those selected for the second batch.

defined as:

$$u_{ent}(x) = -\sum_{i \in [K]} p(y = i|x) \cdot \log p(y = i|x). \tag{5}$$

For feature representation, we take the output of the penultimate layer following CoreSet (Sener & Savarese, 2017) and BADGE (Ash et al., 2019).

**Semantic segmentation**  Semantic segmentation is essentially classification at pixel level. Deep segmentation models output a probability distribution over the label space for each pixel of the input image. To apply our method for semantic segmentation, we first obtain the pixel-level uncertainty and feature vector in a similar way to image classification, and then use mean aggregation to obtain the image-level uncertainty score and feature vector:

$$u(x; \theta_{1:L}) = mean(u(p; \theta_{1:L})), p \in x \tag{6}$$

$$f(x; \theta_{1:L-1}) = mean(f(p; \theta_{1:L-1})), p \in x \tag{7}$$

where $x$ denotes a sample (an image or a patch) and $p$ a pixel within $x$. The aggregated uncertainty and feature are then multiplied to form the embedding using Eq. (4).

**Object detection**  Given an input image, modern object detectors output a set of detections, each associated with a confidence score, a class label and four bounding box coordinates (Ren et al., 2015; Lin et al., 2017b). Previous work (Brust et al., 2018) suggested that the BvSB (Best versus Second Best) metric serves as a better uncertainty measure for object detection task. As our method is not tied to a specific type of loss function or uncertainty measure, this flexibility allows us to select a suitable uncertainty measure for a given task based on prior knowledge. We follow (Brust et al., 2018) to measure the uncertainty of each box by BvSB, which is computed as:

$$u_{bvsb}(x) = \frac{p(y = c^{sb}|x)}{p(y = c^b|x)}, \tag{8}$$

where $c^{sb}$ and $c^b$ is the class label for the second most confident and most confident prediction. The box-level UWE can be obtained in a similar way to image classification. To obtain image-level UWEs, we adopt class-wise aggregation, which is more effective than global aggregation due to the instance-level processing nature of object detection (Agarwal et al., 2020; Wu et al., 2022). Specifically, each class is represented by the UWE of the box with the largest uncertainty within that class:

$$uwe_i(x) = u(\hat{d}_i) \cdot f(\hat{d}_i), \quad i = 1, \ldots, K \tag{9}$$

where $\hat{d}_i = \arg\max_{d \in D_i(x)} u(d)$, $D_i(x)$ is the set of detections with predicted class label $i$ for image $x$, and $K$ is the number of classes. The embedding of each class is then concatenated to form the final representation for the image.

# 4    Experiments

In this section, we report benchmarking results on three different tasks, including image classification, semantic segmentation and object detection. All benchmarking methods start from the same initial randomly selected batch (batch 1). We report the mean and standard deviation of 3 independent runs with different random seeds. To more clearly compare the overall performance under different settings, we summarize the benchmarking results using the pair-wise penalty matrix (PPM) proposed by Ji et al. (2023). More specifically, two AL methods $i$ and $j$ are compared after each batch. If method $i$ outperforms method $j$, a penalty score of $\frac{1}{n}$ (where $n$ is the number of batches) is added to the cell of the PPM at $(i, j)$; likewise, if method $j$ outperforms method $i$, the penalty score is added to the cell at $(j, i)$. The higher the value at $(i, j)$, the stronger the method $i$ dominates method $j$. The column-wise average of PPM (marked as $\Phi$) indicates the overall performance of each method. The method with the lowest value performs the best. The PPMs over various settings (*e.g.*, different datasets, network architectures) can be combined to form a single one by element-wise averaging.

## 4.1    Image Classification

**Datasets**    We evaluate our method on CIFAR-10 (Krizhevsky et al., 2009) and CIFAR-100 datasets (Krizhevsky et al., 2009), each consisting of 50,000 training images and 10,000 testing images of resolution $32 \times 32 \times 3$. We use the training set as the initial unlabelled pool and report the model performance on the testing set.

**Implementation details**    We conduct our experiments on ResNet-18 (He et al., 2016). The hyperparameters for training on both CIFAR-10 and CIFAR-100 are set as follows: batch size = 256, total epochs = 100, initial learning rate = 2e-2 which is decayed by 0.5 after epoch 60 and 80. The fully-supervised baseline is 94.13% accuracy for CIFAR-10 and 75.36% accuracy for CIFAR100.

**Benchmarking results**    We benchmark with Uncertainty (measured by entropy), CoreSet (Sener & Savarese, 2017), BADGE (Ash et al., 2019), BALD (Gal et al., 2017), LLAL (Yoo & Kweon, 2019), VAAL (Sinha et al., 2019), ISAL (Liu et al., 2021) and TA-VAAL (Kim et al., 2021). The results are presented in Fig. 3. We observe that UWE and BADGE perform comparably and outperform the rest. However, the computation of uncertainty-weighted embedding in our method is much more efficient than BADGE's gradient embedding (see Section 5.1 for computational complexity analysis). Single criterion-based method, *e.g.*, Entropy, obtains strong performance on CIFAR-10 and CIFAR-100. This is probably due to the fact that CIFAR-10 and CIFAR-100 are hand-crafted datasets and contains diverse samples by construction. However, single criterion-based method cannot perform well when the dataset characteristic changes, as we shall see in the following experiments on semantic segmentation.

## 4.2    Semantic Segmentation

**Datasets**    We evaluate our method on Cityscapes dataset (Cordts et al., 2016). The dataset contains 19 object classes and the image resolution is $1024 \times 2048$. We use the training set (2,975 images) as the initial unlabelled pool and report the performance on the validation set (500 images). We experiment with both image-level and region-level selection. For region-level AL, each image is divided into patches of size $512 \times 512$, and each patch is considered as a sample for selection and annotation.

**Implementation details**    We conduct our experiments based on the open-source MMSegmentation (Contributors, 2020) framework. The model architecture is DeepLabV3+ with ResNet-18 backbone. The training hyperparameters are set as below: batch size = 8, iterations = 80000, initial learning rate = 1e-2, which is

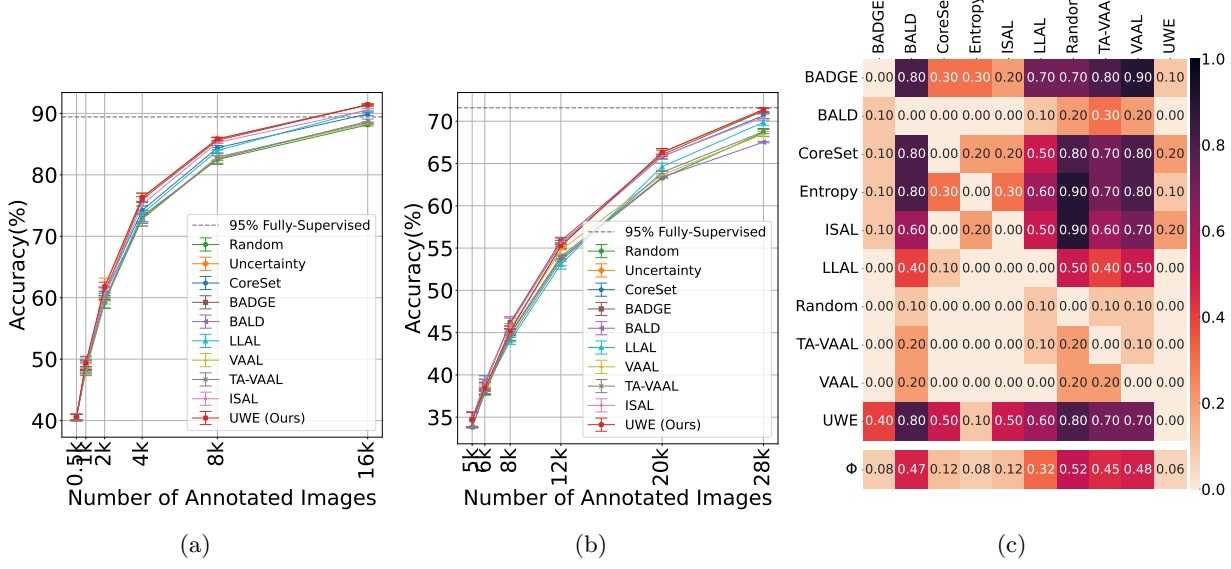

Figure 3: Active learning results of image classification. (a) CIFAR-10; (b) CIFAR-100; (c) PPM over (a) and (b).

decayed using the "poly" policy ($lr = (InitialLR - MinLR) \times (1 - \frac{CurIter}{MaxIter})^{power} + MinLR$, where $MinLR$ = 0.0001 and $power = 0.9$). The fully supervised baseline is 76.30% mIoU.

**Benchmarking results** We benchmark with Uncertainty (measured by entropy), CoreSet (Sener & Savarese, 2017), BADGE (Ash et al., 2019) and CDAL (Agarwal et al., 2020). The results are presented in Fig. 4. We observe that Uncertainty outperforms CoreSet and CDAL (both are diversity-based sampling methods) in image-level AL, but the order is reversed in region-level AL. Region-level AL needs to handle a candidate pool with more redundant samples (as neighboring regions may contain similar content) than image-level AL. Single selection criterion, *e.g.*, uncertainty alone, is unable to accommodate datasets with different degrees of sample redundancy. Our method is able to outperform competing methods in both cases, demonstrating the advantage of the proposed hybrid sampling method in handling datasets of different characteristics.

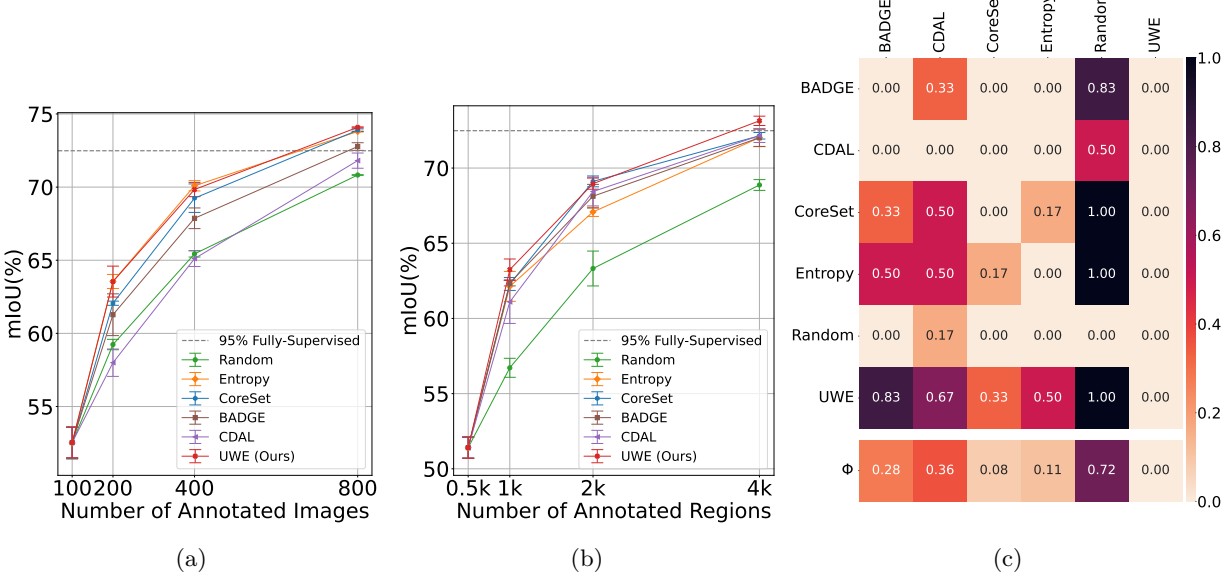

Figure 4: Active learning results of semantic segmentation. (a) Image-level selection; (b) Region-level selection; (c) PPM over (a) and (b).

### 4.3 Object Detection

**Datasets** We evaluate our method on PASCAL VOC0712 dataset (Everingham et al., 2010). There are 20 object classes and the median image shape is $500 \times 375$. Following the setting of previous works (Yoo & Kweon, 2019; Wu et al., 2022), we combine `trainval'07` (5,011 images) and `trainval'12` (11,540 images) to make a super-set `trainval'0712` (16,551 images) and use it as the initial unlabelled pool. The performance is reported on `test'07` (4,952 images).

**Implementation details** Our experiments are based on the open-source MMDetection (Chen et al., 2019) framework. We experiment with one-stage detector RetinaNet (Lin et al., 2017b) as well as two-stage detector Faster R-CNN (Ren et al., 2015). Both detectors use feature pyramid networks (Lin et al., 2017a) on top of ResNet-50 (He et al., 2016) as feature extractor. The hyperparameters are set as follows: batch size = 2, total epochs = 26, initial learning rate = 1e-3 for RetinaNet and 5e-3 for Faster R-CNN, which is decayed by 0.1 after 20 epochs. When training with the full `trainval'0712`, we obtained a mAP of 79.87% for RetinaNet and 78.73% for Faster R-CNN.

**Benchmarking results** We benchmark with Uncertainty (measured by BvSB), CoreSet (Sener & Savarese, 2017), BADGE (Ash et al., 2019), CDAL (Agarwal et al., 2020), LLAL (Yoo & Kweon, 2019), EnmsDivproto (Wu et al., 2022) and CALD (Yu et al., 2022). The results are presented in Fig. 5. We observe that not every method can perform well on both Faster R-CNN and RetinaNet, *e.g.*, CALD and BADGE perform well with the former, but the performance suffers with the latter. UWE is able to consistently perform the best on both detectors. EnmsDivproto also achieves strong performance on both detectors, but the method is more complicated, involving additional steps for entropy-based non-maximum suppression and pseudo-label based minority class sampling on top of diversity-based sampling on class prototypes, while our method is simple and efficient (up to 4× faster than EnmsDivproto in selection), as it only involves diversity-based sampling on class-wise aggregated features.

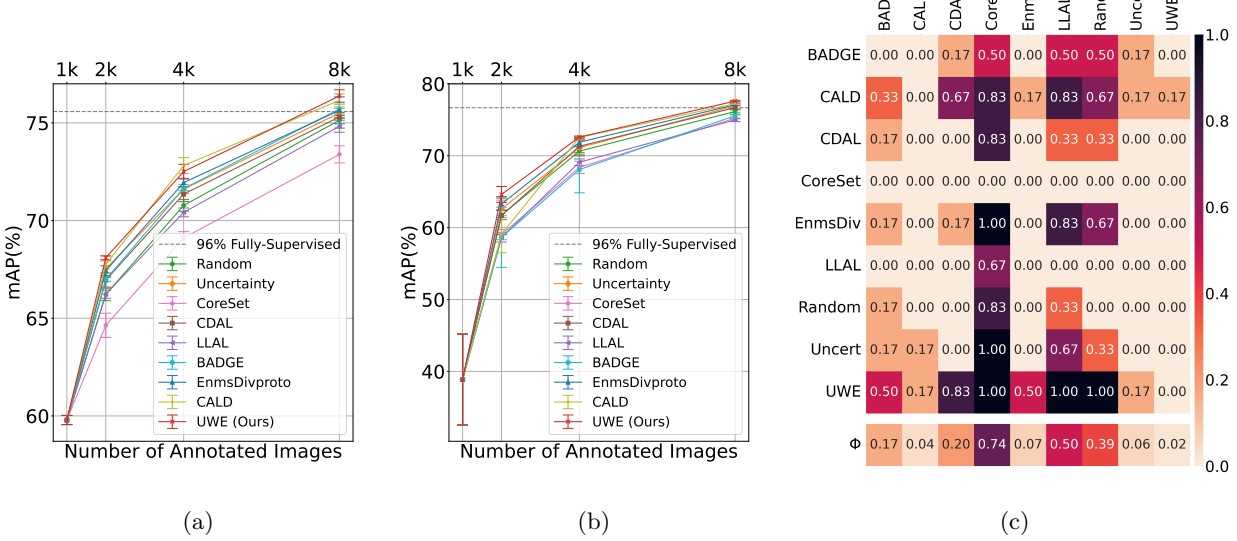

Figure 5: Active learning results of object detection. (a) Faster R-CNN; (b) RetinaNet; (c) PPM over (a) and (b).

### 4.4 Ablation Studies

We report ablation studies on the design components of our method in Table 1. Here, U represents uncertainty sampling that selects the top-k most uncertain samples, F represents diversity sampling using KCG without uncertainty weighting, *i.e.*, setting $u(x)$ in Eq. (4) to 1, and U+F refers to the proposed method. The consistent performance gain of U+F over F demonstrates the advantage of performing diversity sampling on the proposed uncertainty-weighted features.

Table 1: Ablation studies on the design choices of our method. U represents uncertainty sampling that selects the top-k most uncertain samples, F represents diversity sampling using KCG without uncertainty weighting, and U+F is the proposed method.

| U | F | Batch 2 | Batch 3 | Batch 4 |
|---|---|---|---|---|
| colspan Image classification on CIFAR10 with ResNet-18 | | | | |
| ✓ | | 46.96 (0.29) | 60.08 (0.08) | 76.02 (0.20) |
| | ✓ | 48.61 (0.71) | 60.46 (0.14) | 74.35 (0.41) |
| ✓ | ✓ | **49.45** (0.41) | **61.84** (0.37) | **76.37** (0.99) |
| Semantic segmentation on Cityscapes with DeepLabV3+ | | | | |
| ✓ | | 61.63 (0.85) | 66.88 (0.07) | 71.77 (0.59) |
| | ✓ | 61.82 (1.04) | 68.19 (1.25) | 71.75 (0.57) |
| ✓ | ✓ | **62.78** (0.98) | **68.42** (0.13) | **72.87** (0.71) |
| Object detection on VOC0712 with Faster R-CNN | | | | |
| ✓ | | 67.53 (0.40) | 71.61 (0.23) | 75.51 (0.23) |
| | ✓ | 67.20 (0.29) | 71.75 (0.38) | 75.83 (0.51) |
| ✓ | ✓ | **68.09** (0.10) | **72.51** (0.37) | **76.38** (0.32) |

## 5 Further Analysis

### 5.1 Computational Complexity Analysis

We analyze the computational complexity of our method to demonstrate its efficiency over BADGE. Let $f_{dim}$ denote the dimension of the feature representation, and $n_c$ the number of classes. For image classification and segmentation, the dimension of UWE is the same as $f_{dim}$, while the dimension of the gradient embedding used in BADGE is $n_c \cdot f_{dim}$. The time complexity of KCG and KM++ is $O(n_b \cdot n_u \cdot n_{dim})$, where $n_b$ is the number of selected samples, $n_u$ is the total number of unlabelled samples, and $n_{dim}$ is the input sample dimension. As such, our method is more efficient than BADGE by a factor of $n_c$. For object detection, the dimension of UWE is $n_c \cdot f_{dim}$ due to class-wise aggregation. The dimension of gradient embedding varies according to detector architectures and is typically much higher as it not only depends on the number of classes, but also other network parameters. For example, the dimension of gradient embedding for RetinaNet is $k^2 \cdot n_a \cdot n_c \cdot f_{dim}$, where $n_a$ is the number of anchors and $k$ is the filter size for the prediction head. Table 2 compares the embedding dimension, embedding extraction time and sample selection time on BADGE *vs.* UWE. Our method reduces the storage and time cost significantly compared to BADGE, especially when the number of classes is large (*i.e.*, on CIFAR-100) and for dense prediction task (*i.e.*, on Cityscapes).

Table 2: The dimension of the embedding and running time (in seconds) for embedding extraction and sample selection. The time cost is measured for selecting the second batch.

| | Dimension | Extraction | Selection |
|---|---|---|---|
| Image classification on CIFAR-10 with ResNet-18 | | | |
| BADGE | 5120 | 18.84 (1.30) | 129.95 (1.23) |
| UWE | **512** | **4.78** (0.05) | **22.55** (0.28) |
| Image classification on CIFAR-100 with ResNet-18 | | | |
| BADGE | 51200 | 126.87 (12.20) | 2223.94 (0.18) |
| UWE | **512** | **4.98** (0.03) | **42.16** (0.08) |
| Semantic segmentation on Cityscapes with DeepLabV3+ | | | |
| BADGE | 2432 | 6593.17 (618.16) | 34.18 (5.27) |
| UWE | **128** | **997.26** (12.13) | **23.22** (0.34) |
| Object detection on VOC0712 with RetinaNet | | | |
| BADGE | 414720 | 1651.34 (21.22) | 200.66 (30.87) |
| UWE | **5120** | **1181.62** (166.13) | **82.33** (2.62) |

## 5.2 Adjusting Trade-off between Uncertainty and Diversity Sampling

In Section 4.2, we observe that uncertainty sampling outperforms diversity sampling in image-level AL, but in region-level AL the order is reversed. Intuitively, when region size is large, candidate samples are already diverse and uncertainty sampling that focuses on selecting difficult samples would outperform. On the other hand, when region size is small, there are more redundant samples, and thus a selection strategy that encourages diversity sampling would outperform. This suggests that the trade-off between uncertainty and diversity sampling needs to be adjusted for different region sizes. This can be achieved by applying a power $\tau$ to the uncertainty term in Eq. (4), *i.e.*, computing the embedding as $uwe(x) = u(x)^\tau \cdot f(x)$. Increasing $\tau$ would up-weight the contribution of uncertainty and select more uncertain samples, while decreasing $\tau$ would down-weight the role of uncertainty and select more diverse samples. In Section 4.2, we show that $\tau = 1$ works well for $1024 \times 2048$ and $512 \times 512$. If the region size is further reduced, a smaller $\tau$ should be more suitable. To verify this, we experiment with smaller region sizes using reduced $\tau$. Results are reported in Table 3. When region size is $256 \times 256$, we observe some marginal advantage of UWE ($\tau = 0.1$) over UWE ($\tau = 1$). The advantage is more significant when the region size is further reduced to $128 \times 128$. We observe that CoreSet significantly outperforms Entropy at this region size, confirming the advantage of diversity sampling for datasets with more redundant samples. UWE with $\tau = 0.1$ can further improve upon CoreSet, demonstrating that a hybrid approach that considers uncertainty in diversity sampling is still beneficial.

Table 3: Effect of applying power $\tau$ to the uncertainty term in computing the embedding.

|  | Batch 2 | Batch 3 | Batch 4 |
|---|---|---|---|
| Region size $= 256 \times 256$ | | | |
| Random | 60.35 (0.17) | 64.75 (0.38) | 68.70 (0.59) |
| Entropy | 67.18 (0.52) | 71.41 (0.13) | 72.67 (0.68) |
| CoreSet | 66.38 (0.08) | 71.19 (0.36) | 72.86 (0.31) |
| BADGE | 65.10 (0.21) | 69.87 (0.14) | 70.03 (1.15) |
| UWE ($\tau = 1$) | 67.21 (0.46) | 71.21 (0.80) | **73.72** (0.24) |
| UWE ($\tau = 0.1$) | **67.23** (0.21) | **71.65** (0.67) | 73.71 (0.18) |
| Region size $= 128 \times 128$ | | | |
| Random | 62.56 (0.09) | 67.38 (0.29) | 69.05 (1.70) |
| Entropy | 70.09 (0.33) | 69.99 (0.43) | 71.38 (0.09) |
| CoreSet | 70.41 (0.34) | 72.49 (0.54) | 74.28 (0.47) |
| BADGE | 68.50 (0.00) | 71.91 (0.00) | 71.78 (0.00) |
| UWE ($\tau = 1$) | 69.80 (0.26) | 72.10 (0.59) | 73.40 (0.13) |
| UWE ($\tau = 0.1$) | **70.75** (0.49) | **73.15** (0.24) | **74.40** (0.37) |

## 5.3 Effect of Uncertainty Measures

The $u(x)$ term in Eq. (4) can be an arbitrary measure of uncertainty. We experiment with three commonly used measures: Entropy, BvSB and LC. Figure 6 shows the results. While the performance of different uncertainty measures varies across different datasets and tasks, our method is able to consistently improve upon the performance of uncertainty sampling. This demonstrates the effectiveness of UWE as a general method to perform hybrid AL.

## 5.4 Correlation between Gradient Embedding and UWE

We inspect the correlation between the magnitude of gradient embedding and UWE to shed more insight into the proposed two-term decomposition of gradient embedding. Mathematically, the magnitude of gradient embedding under cross-entropy loss and softmax activation function can be computed as:

$$||ge(x, \hat{y})|| = (\sum_i^K p_i^2 + 1 - 2p_{\hat{y}})^{1/2} \cdot ||f(x)||, \tag{10}$$

where $p_{\hat{y}}$ is the probability for the pseudo label. The magnitude of UWE can be computed as:

$$||uwe(x)|| = u(x) \cdot ||f(x)||. \tag{11}$$

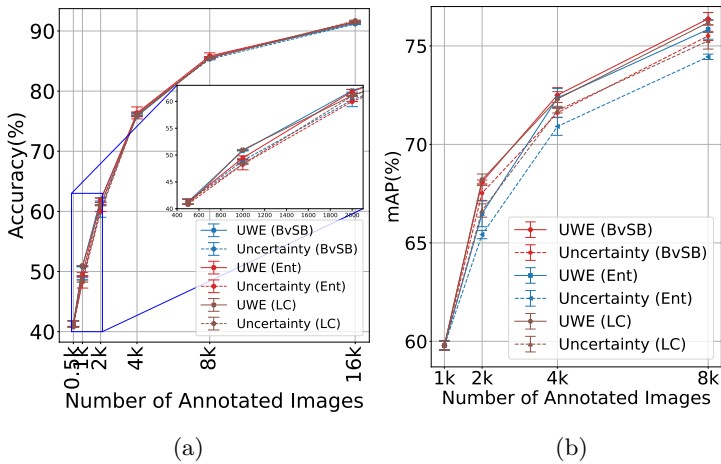

(a)  (b)

Figure 6: Performance of UWE with different uncertainty measures. (a) Image classification on CIFAR-10 with ResNet-18; (b) Object detection on VOC0712 with Faster R-CNN.

It is clear that $||ge(x, \hat{y})||$ and $||uwe(x)||$ share the same second term $||f(x)||$ (the feature representation), while the correlation between the first terms depends on the choice of uncertainty measure and the joint distribution $p(x, y)$. We compute the Pearson correlation coefficient between $||ge(x, \hat{y})||$ and $||uwe(x)||$. Pearson correlation coefficient measures the linear correlation between two sets of data, with 1/-1 indicating perfect positive/negative correlation, and 0 indicating no linear correlation. Results are reported in Table 4. We observe that the magnitude of BADGE's gradient embedding and UWE is highly correlated, achieving correlation coefficient above 0.9. We further inspect how well BADGE and UWE in approximating real gradient, *i.e.*, gradient embedding computed with ground truth label. Results in Table 5 show that UWE obtains similar, or sometimes higher correlation coefficients than BADGE, suggesting that UWE performs comparably to BADGE with regard to approximating the magnitude of real gradient.

Table 4: Correlation between the magnitude of BADGE's gradient embedding and UWE. We report the mean and standard deviation of 3 runs.

|  | Batch 1 | Batch 2 | Batch 3 | Batch 4 | Batch 5 | Batch 6 |
|---|---|---|---|---|---|---|
| CIFAR-10 | 0.94 (0.00) | 0.94 (0.00) | 0.95 (0.00) | 0.95 (0.00) | 0.95 (0.00) | 0.95 (0.00) |
| CIFAR-100 | 0.93 (0.00) | 0.93 (0.00) | 0.94 (0.01) | 0.93 (0.00) | 0.93 (0.00) | 0.94 (0.00) |

Table 5: Correlation between the magnitude of real gradient (using ground truth label) and the magnitude of BADGE's gradient embedding (using pseudo label) and UWE, respectively. We report the mean and standard deviation of 3 runs.

|  | Batch 1 | Batch 2 | Batch 3 | Batch 4 | Batch 5 | Batch 6 |
|---|---|---|---|---|---|---|
| Image classification on CIFAR-10 with ResNet-18 | | | | | | |
| BADGE | 0.11 (0.00) | 0.19 (0.02) | 0.28 (0.00) | 0.37 (0.00) | 0.44 (0.01) | 0.51 (0.00) |
| UWE | 0.12 (0.00) | 0.22 (0.02) | 0.32 (0.01) | 0.42 (0.00) | 0.48 (0.01) | 0.55 (0.00) |
| Image classification on CIFAR-100 with ResNet-18 | | | | | | |
| BADGE | 0.46 (0.00) | 0.47 (0.03) | 0.55 (0.00) | 0.59 (0.04) | 0.61 (0.01) | 0.67 (0.02) |
| UWE | 0.47 (0.00) | 0.48 (0.03) | 0.56 (0.00) | 0.59 (0.04) | 0.61 (0.01) | 0.65 (0.03) |

## 6 Conclusions

In this work, we proposed a novel uncertainty-weighted embedding for hybrid AL. Our proposed uncertainty-weighted embedding generalizes the gradient embedding form of BADGE by allowing arbitrary uncertainty measures to be used, thereby greatly reducing the computational complexity of computing the embedding,

and allowing it to be used with arbitrary loss functions. Our hybrid AL method then utilizes this embedding as a basis for distance-based sampling to select examples for labelling. We evaluate our method under a wide range of tasks and settings, *e.g.*, image classification, image-level and region-level AL for semantic segmentation, one-stage and two-stage detectors for object detection. Experimental results show that our method achieves state-of-the-art performance, and demonstrate the generality of our approach.

## Acknowledgements

The authors would like to thank the anonymous TMLR reviewers for their constructive feedback during the review process. This work is supported by the Agency for Science, Technology and Research (A*STAR) under its AME Programmatic Funds (Grant No. A20H6b0151) and Career Development Fund (Grant No. C210812052). The computational work for this article was partially performed on resources of the National Supercomputing Centre, Singapore (https://www.nscc.sg).

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

## A  Appendix

### A.1  Effect of Selection Methods

We compare the uncertainty and diversity of samples selected by KCG and KM++ on both UWE and gradient embedding. Diversity is measured by the average pairwise distance of samples within a batch. Results are presented in Fig. 7. KCG selects samples of larger uncertainty, embedding magnitude and diversity on both embeddings. We also follow BADGE to measure diversity by the log determinant of the selected samples' Gram matrix in Table 6. We observe similar results to those reported in Fig. 7, *i.e.*, KCG is able to select a more diverse batch than KM++. We further experiment with replacing KM++ used in

BADGE by KCG. The results presented in Fig. 8 demonstrate that KCG performs better than KM++ on gradient embeddings as well. One of the weaknesses of KCG is that it may select outliers that are far away from each other (Sener & Savarese, 2017). However, public benchmarking datasets are usually carefully crafted and may not suffer from the outlier problem. As a result, the deterministic nature of KCG allows it to select more diverse and uncertain samples and performs better than KM++.

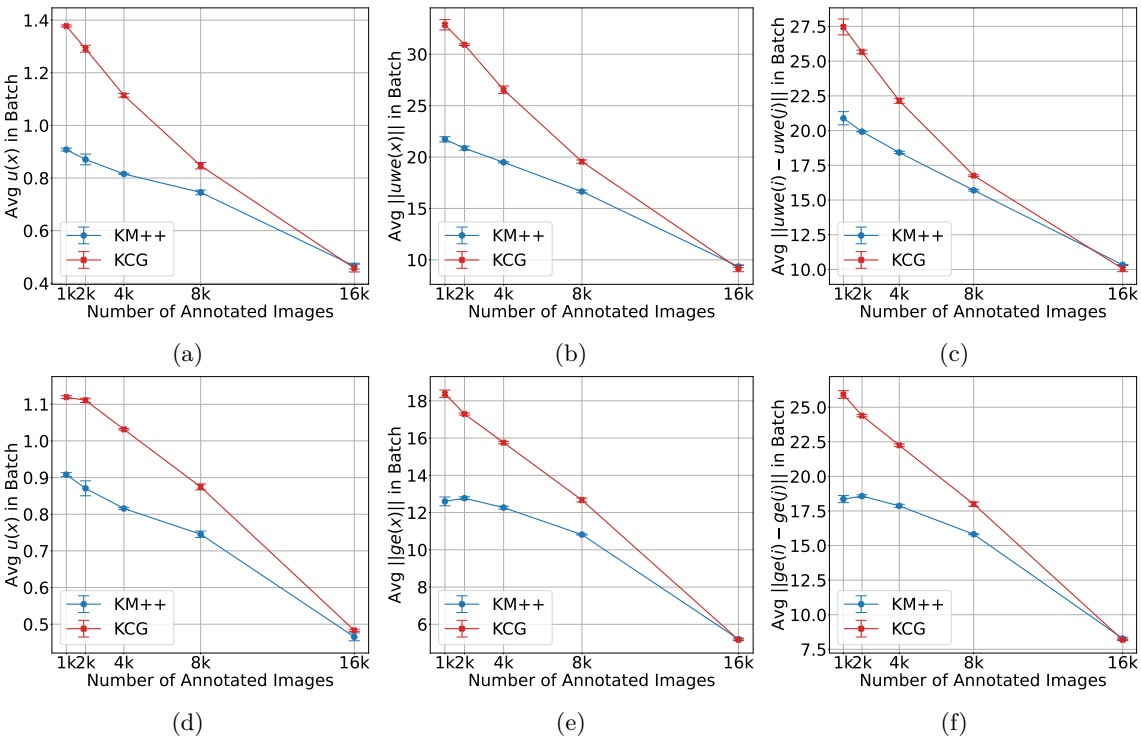

Figure 7: Comparing uncertainty and diversity of samples selected by KCG and KM++ on CIFAR-10. Top row: results on uncertainty-weighted embeddings; bottom row: results on gradient embeddings. (a)(d): Comparing selected samples' uncertainty; (b)(e): Comparing selected samples' embedding magnitude; (c)(f): Comparing selected samples' diversity (measured by average pairwise feature distance within a batch).

Table 6: The log determinant of Gram matrix of samples selected by KCG *vs.* KM++ on CIFAR-10. We compute the Gram matrix on both the selected samples' UWEs and their original (unweighted) features. KCG obtains larger determinant values in both cases, indicating the selected samples are more diverse.

| Log determinant of Gram matrix of UWEs in batch | | | |
|---|---|---|---|
| Batch (Budget) | 2 (500) | 3 (1000) | 4 (2000) |
| KM++ | 813.86 (5.04) | -11789.39 (70.67) | -37898.37 (7.50) |
| KCG | **1353.61** (4.02) | **-10780.49** (18.76) | **-36313.16** (70.82) |
| Log determinant of Gram matrix of (unweighted) features in batch | | | |
| KM++ | 1034.34 (0.38) | -1704.00 (3.20) | -7499.03 (0.48) |
| KCG | **1088.14** (0.31) | **-1643.87** (3.35) | **-7470.63** (25.94) |

## A.2 Effect of Aggregation for Object Detection

We use *Max* aggregation to generate class-wise UWE (Eq. (9)) for object detection. Alternatively, the class-wise UWE can be obtained by averaging the box-level UWE within each class, *i.e.*, $uwe_i(x) = \frac{\sum_{d \in D_i(x)} u(d) \cdot f(d)}{|D_i(x)|}$, where $D_i(x)$ is the set of detections with predicted class label $i$ for image $x$. Table 7

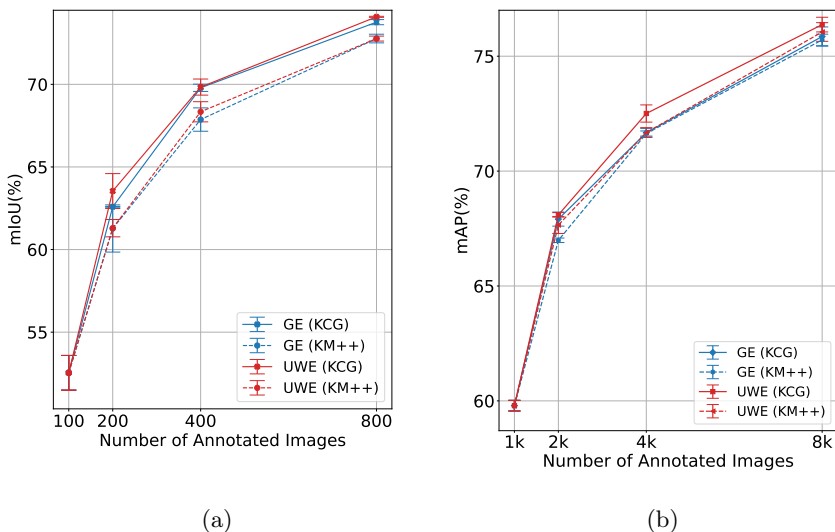

Figure 8: Replacing KM++ in BADGE with KCG. KCG performs better than KM++ on both gradient embeddings and uncertainty-weighted embeddings. (a) Semantic segmentation on Cityscapes; (b) Object detection on VOC0712 with Faster R-CNN.

compares the performance of *Max vs. Mean* aggregation. We observe that *Max* performs better or comparably to *Mean*. The reason may be that *Max* can generate a more distinctive embedding as each class is represented by its most uncertain object instead of its "average" object, and more distinctive embeddings facilitate the selection of more informative samples by distance-based sampling.

Table 7: Comparing *Max vs. Mean* aggregation for generating class-wise UWE for object detection.

|  | Batch 2 | Batch 3 | Batch 4 |
|---|---|---|---|
| Object detection on VOC0712 with Faster R-CNN | | | |
| *Mean* | 67.18 (0.66) | 72.03 (0.39) | 75.69 (0.19) |
| *Max* | **68.09** (0.09) | **72.50** (0.37) | **76.37** (0.31) |
| Object detection on VOC0712 with RetinaNet | | | |
| *Mean* | 64.08 (0.38) | **72.79** (0.09) | **77.62** (0.20) |
| *Max* | **64.61** (1.10) | 72.58 (0.12) | 77.59 (0.10) |

## A.3 BADGE for Object Detection

To evaluate BADGE's effectiveness for object detection, we experiment with variants of BADGE. Modern object detectors like RetinaNet (Lin et al., 2017b) and Faster R-CNN (Ren et al., 2015) typically consists of two prediction branches, *i.e.*, classification (CLS) and regression (REG). Two-stage detector like Faster R-CNN has another prediction branch for region proposal (RPN, which itself consists of a classification and regression head). We experiments with three variants of BADGE, *i.e.*, BADGE (CLS), BADGE (CLS+REG) and BADGE (CLS+REG+RPG), where the last layer parameters of the corresponding branches are taken into consideration when computing the gradient embedding. We observe that BADGE (CLS) performs the best, while using the gradient embedding from the regression branch or region proposal network harms the performance. These results suggest that BADGE is not guaranteed to work well for complicated tasks like object detection.

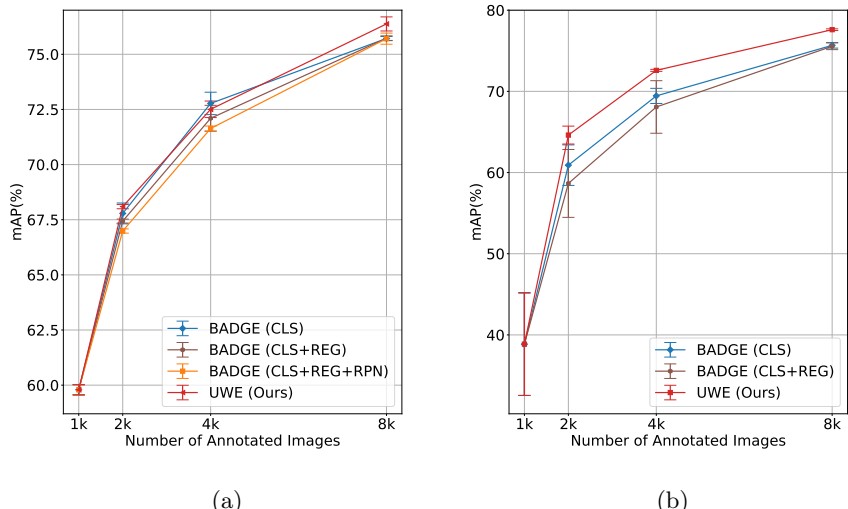

(a)         (b)

Figure 9: BADGE for object detection with variants of gradient embeddings. (a) Results on VOC0712 with Faster R-CNN; (b) Results on VOC0712 with RetinaNet.

### A.4 Uncertainty and Diversity of Selected Samples

To validate our method is indeed able to select a set of both uncertain and diverse samples, we compare the mean uncertainty and diversity (measured by average pairwise feature distance within a batch) of samples selected by different methods in Fig. 10. We observe that UWE obtains the second best regarding to uncertainty, just behind Uncertainty sampling, which selects samples of top-k uncertainty. However, Uncertainty sampling does not consider diversity, thus ranking bottom regarding to diversity for image classification. Our method, being a hybrid method, is still able to achieve good diversity for image classification and ranks first for object detection.

### A.5 Results on mini-Imagenet

We present the results on mini-Imagenet (Vinyals et al., 2016) in Fig. 11. Mini-Imagenet contains 48,000 images for training and 12,000 images for testing, evenly distributed across 100 classes. The image resolution is $84 \times 84 \times 3$. We use the training set as the initial unlabelled pool and the testing set to evaluate model performance. The network architecture is Vision Transformers (ViT-Base) (Dosovitskiy et al., 2020), which is initialized with ImageNet pre-trained weights (self-supervised pre-training with DINO (Caron et al., 2021)). At each AL cycle, the model is trained for 20 epochs by ADAM optimizer with learning rate 5e-5 and batch size 64. The fully-supervised baseline achieves 80.60% in testing accuracy. We use BvSB as uncertainty measure for UWE. We observe that UWE performs slightly better than BADGE, and outperform others significantly.

### A.6 Qualitative Results

We provide detailed detection results of models trained with samples selected by different active learning methods in Fig. 12. Our method is able to obtain more accurate detection results, with fewer false positives (*e.g.*, dog/cat in Fig. 12(b)) and false negatives (*e.g.*, potted plant in Fig. 12(d)).

### A.7 Numerical Results

We provide the raw data for Figs.3, 4 and 5 in Table 8, 9 and 10, respectively. Our method (UWE) achieves best or second best performance in most cases.

| Batch (Budget) | 1 (500) | 2 (500) | 3 (1000) | 4 (2000) | 5 (4000) | 6 (8000) |
|---|---|---|---|---|---|---|
| ResNet-18 on CIFAR-10 | | | | | | |
| Random | 41.3 (0.49) | 49.97 (1.16) | 60.31 (0.11) | 72.78 (0.17) | 82.42 (0.66) | 88.21 (0.04) |
| Uncertainty | 41.3 (0.49) | 46.96 (0.29) | 60.08 (0.09) | 76.02 (0.20) | **85.96** (0.19) | **91.60** (0.07) |
| CoreSet | 41.3 (0.49) | 48.61 (0.71) | 60.46 (0.14) | 74.35 (0.41) | 84.11 (0.28) | 90.11 (0.05) |
| BADGE | 41.3 (0.49) | 48.35 (0.51) | 61.56 (1.02) | 75.75 (0.52) | 85.64 (0.42) | 91.40 (0.22) |
| BALD | 41.3 (0.49) | **50.09** (0.65) | 59.68 (0.27) | 73.63 (0.31) | 82.99 (0.58) | 88.58 (0.25) |
| LLAL | 41.3 (0.49) | 48.73 (0.68) | 59.89 (0.23) | 74.37 (0.23) | 84.77 (0.43) | 90.50 (0.12) |
| VAAL | 41.3 (0.49) | 48.26 (0.72) | 59.09 (0.13) | 72.62 (0.58) | 82.99 (0.32) | 88.39 (0.10) |
| TA-VAAL | 41.3 (0.49) | 48.53 (1.23) | 59.45 (1.21) | 72.90 (1.27) | 82.63 (0.96) | 88.71 (0.37) |
| ISAL | 41.3 (0.49) | 49.31 (0.90) | 60.59 (1.12) | 75.15 (1.25) | 85.26 (0.17) | 90.55 (0.25) |
| UWE (Ours) | 41.3 (0.49) | 49.46 (0.41) | **61.84** (0.37) | **76.38** (1.00) | 85.85 (0.51) | 91.46 (0.08) |

| Batch (Budget) | 1 (5000) | 2 (1000) | 3 (2000) | 4 (4000) | 5 (8000) | 6 (8000) |
|---|---|---|---|---|---|---|
| ResNet-18 on CIFAR-100 | | | | | | |
| Random | 33.83 (0.08) | 37.87 (0.28) | 44.79 (0.71) | 53.77 (0.30) | 63.28 (0.05) | 68.68 (0.46) |
| Uncertainty | 33.83 (0.08) | 36.98 (0.39) | **45.99** (0.14) | 55.66 (0.58) | 66.23 (0.03) | 71.41 (0.11) |
| CoreSet | 33.83 (0.08) | 38.54 (0.59) | 45.45 (1.57) | 55.52 (0.35) | 65.63 (0.10) | 70.62 (0.20) |
| BADGE | 33.83 (0.08) | 38.20 (0.25) | 45.98 (0.38) | **55.69** (0.07) | 66.51 (0.03) | 70.99 (0.03) |
| BALD | 33.83 (0.08) | 38.10 (0.06) | 44.93 (0.16) | 54.02 (0.27) | 63.39 (0.13) | 67.53 (0.09) |
| LLAL | 33.83 (0.08) | 37.54 (0.28) | 43.09 (1.05) | 52.79 (0.03) | 65.09 (0.41) | 69.97 (0.16) |
| VAAL | 33.83 (0.08) | 38.73 (0.30) | 44.84 (0.36) | 53.45 (0.14) | 63.35 (0.47) | 68.74 (0.15) |
| TA-VAAL | 33.83 (0.08) | 38.07 (0.39) | 44.37 (0.46) | 53.50 (0.58) | 63.96 (0.17) | 68.77 (0.24) |
| ISAL | 33.83 (0.08) | **39.17** (0.36) | 45.92 (0.98) | 55.57 (0.55) | 65.86 (0.28) | 70.46 (0.36) |
| UWE (Ours) | 33.83 (0.08) | 38.49 (0.55) | 45.28 (0.08) | 55.59 (0.80) | **67.11** (0.48) | **72.09** (0.12) |

Table 8: Benchmarking results for image classification. Best results are bolded, and the second best results are underlined.

| Batch (Budget) | 1 (100) | 2 (100) | 3 (200) | 4 (400) |
|---|---|---|---|---|
| DeepLabv3+ on Cityscapes (Image-level AL) | | | | |
| Random | 52.54 (1.05) | 59.24 (0.36) | 65.44 (0.22) | 70.83 (0.02) |
| Uncertainty | 52.54 (1.05) | **63.55** (0.48) | **70.09** (0.35) | 73.80 (0.03) |
| CoreSet | 52.54 (1.05) | 62.06 (0.14) | 69.24 (0.97) | 73.89 (0.09) |
| CDAL | 52.54 (1.05) | 58.00 (0.94) | 65.11 (0.54) | 71.81 (0.52) |
| BADGE | 52.54 (1.05) | 61.28 (1.43) | 67.87 (0.71) | 72.77 (0.26) |
| UWE (Ours) | 52.54 (1.05) | 63.54 (1.06) | 69.83 (0.49) | **74.09** (0.03) |

| Batch (Budget) | 1 (500) | 2 (500) | 3 (1000) | 4 (2000) |
|---|---|---|---|---|
| DeepLabv3+ on Cityscapes (Region-level AL with region size $512 \times 512$) | | | | |
| Random | 50.71 (1.19) | 56.94 (0.67) | 63.73 (1.23) | 69.00 (0.38) |
| Uncertainty | 50.71 (1.19) | 61.63 (0.85) | 66.88 (0.07) | 71.77 (0.59) |
| CoreSet | 50.71 (1.19) | 61.82 (1.04) | 68.19 (1.25) | 71.75 (0.57) |
| CDAL | 50.71 (1.19) | 61.01 (0.11) | 68.03 (0.77) | 72.75 (1.09) |
| BADGE | 50.71 (1.19) | 61.60 (0.83) | 67.07 (0.16) | 71.24 (0.04) |
| UWE (Ours) | 50.71 (1.19) | **62.78** (0.98) | **68.42** (0.13) | **72.87** (0.71) |

Table 9: Benchmarking results for semantic segmentation. Best results are bolded, and the second best results are underlined.

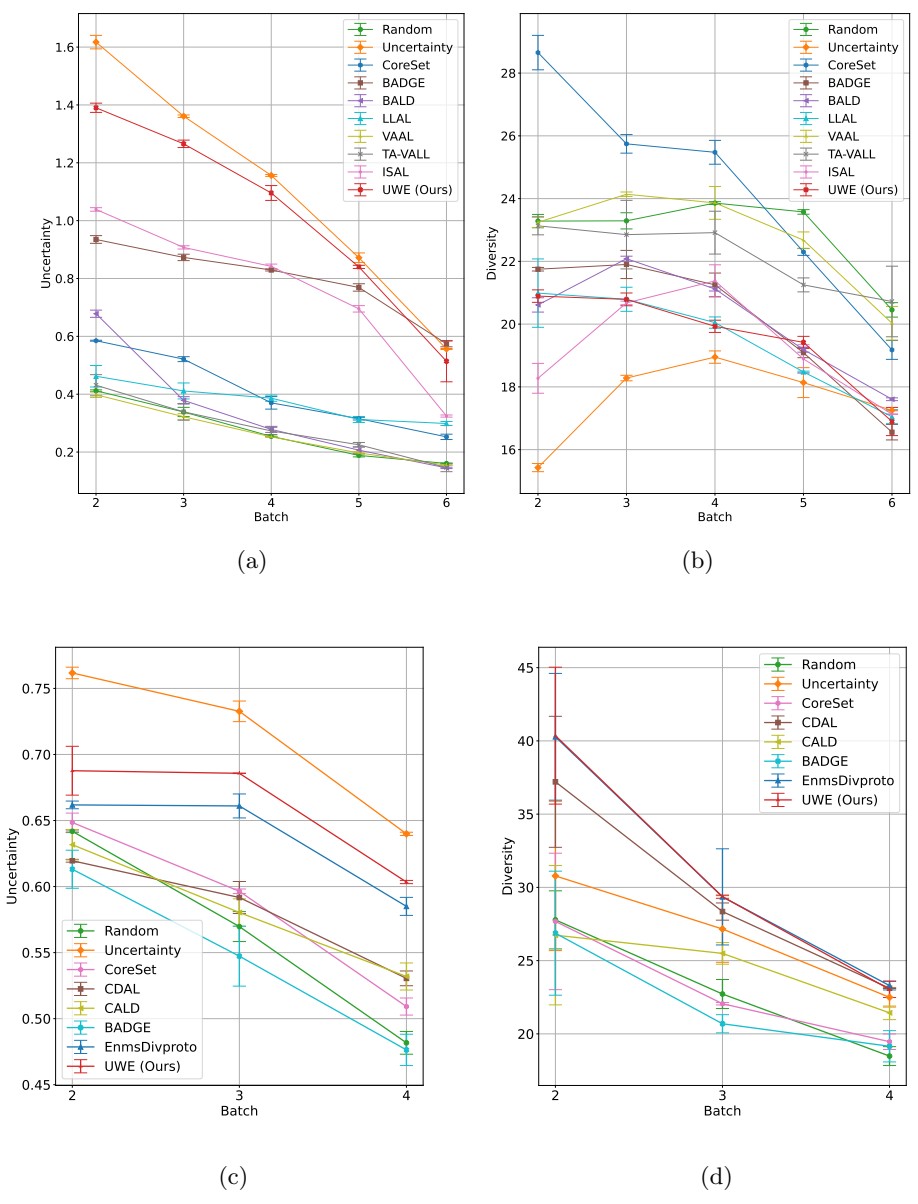

Figure 10: Comparing uncertainty and diversity of samples selected by different AL methods. (a)(b): Image classification on CIFAR-10 with ResNet-18; (c)(d): Object detection on VOC0712 with RetinaNet.

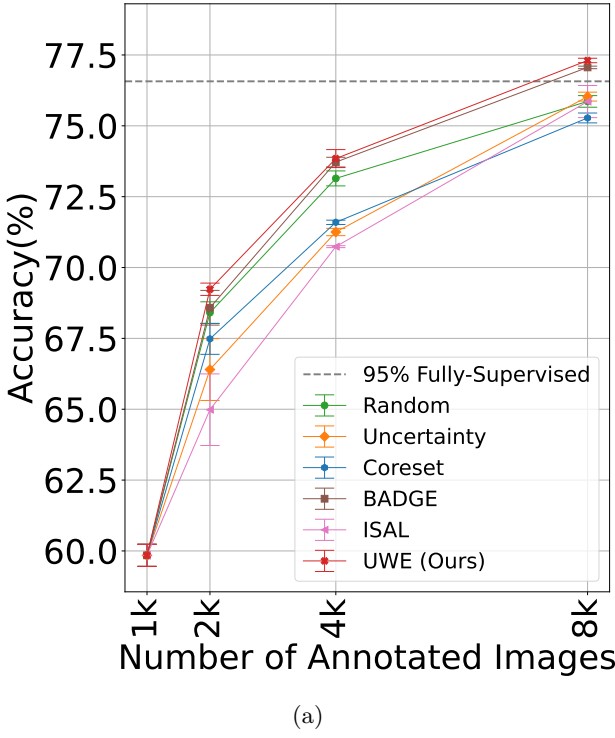

(a)

Figure 11: Active learning results on mini-Imagenet.

| Batch (Budget) | 1 (1000) | 2 (1000) | 3 (2000) | 4 (4000) |
|---|---|---|---|---|
| Faster R-CNN on VOC0712 | | | | |
| Random | 59.79 (0.23) | 66.22 (0.32) | 70.78 (0.30) | 75.14 (0.40) |
| Uncertainty | 59.79 (0.23) | 67.53 (0.40) | 71.61 (0.23) | 75.51 (0.23) |
| CoreSet | 59.79 (0.23) | 64.64 (0.62) | 69.11 (0.32) | 73.39 (0.44) |
| CDAL | 59.79 (0.23) | 66.95 (0.36) | 71.35 (0.38) | 75.31 (0.07) |
| LLAL | 59.79 (0.23) | 66.23 (0.21) | 70.42 (0.23) | 74.83 (0.31) |
| BADGE | 59.79 (0.23) | 66.98 (0.09) | 71.63 (0.11) | 75.70 (0.25) |
| EnmsDivproto | 59.79 (0.23) | 67.44 (0.25) | 71.94 (0.22) | 75.64 (0.69) |
| CALD | 59.79 (0.23) | 67.75 (0.43) | **72.81** (0.40) | 76.15 (0.33) |
| UWE (Ours) | 59.79 (0.23) | **68.10** (0.10) | 72.51 (0.37) | **76.38** (0.32) |
| RetinaNet on VOC0712 | | | | |
| Random | 38.86 (6.32) | 61.75 (0.63) | 70.64 (0.67) | 76.14 (0.38) |
| Uncertainty | 38.86 (6.32) | 62.61 (1.21) | 71.12 (0.96) | 76.93 (0.40) |
| CoreSet | 38.86 (6.32) | 59.02 (0.70) | 68.37 (0.86) | 75.21 (0.45) |
| CDAL | 38.86 (6.32) | 61.68 (2.63) | 71.32 (0.90) | 76.69 (0.31) |
| LLAL | 38.86 (6.32) | 58.63 (0.64) | 69.10 (0.46) | 74.98 (0.23) |
| BADGE | 38.86 (6.32) | 58.65 (4.18) | 68.07 (3.23) | 75.55 (0.41) |
| EnmsDivproto | 38.86 (6.32) | 63.29 (0.83) | 71.87 (0.62) | 77.09 (0.11) |
| CALD | 38.86 (6.32) | 57.50 (0.20) | 68.64 (0.50) | 76.18 (0.45) |
| UWE (Ours) | 38.86 (6.32) | **64.62** (1.10) | **72.59** (0.12) | **77.60** (0.10) |

Table 10: Benchmarking results for object detection. Best results are bolded, and the second best results are underlined.

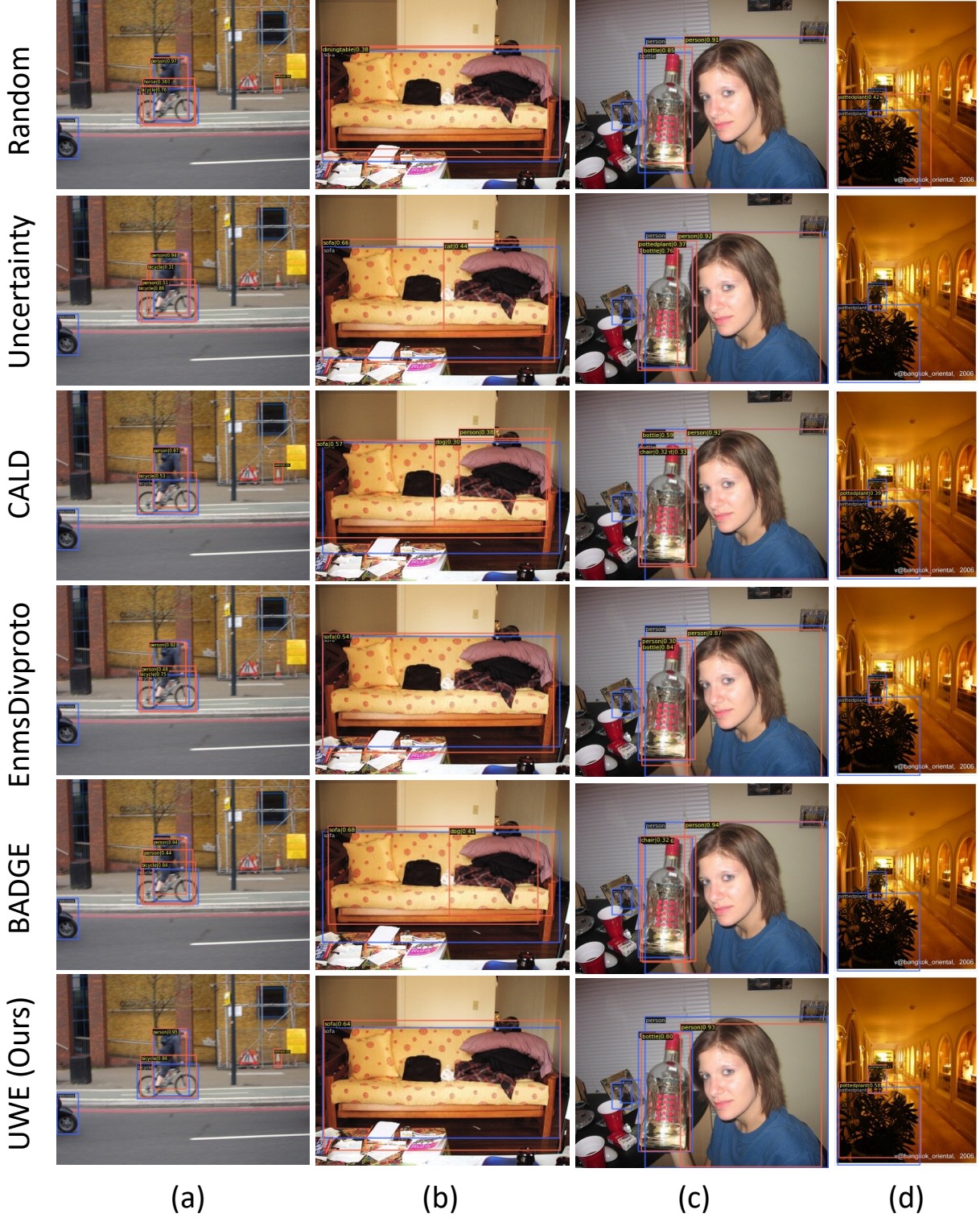

Figure 12: Visualization of detection results of RetinaNet on VOC0712. Blue boxes denote ground truth and orange boxes denote detection results. Model is trained with 2k images selected by various active learning methods. Our method is able to obtain detection results of higher quality, with fewer false positives (*e.g.*, dog/cat in (b)) and false negatives (*e.g.*, potted plant in (d)).

