# OpenReview forum: "Hybrid Active Learning with Uncertainty-Weighted Embeddings"
_TMLR — Accepted by TMLR_

### Review · Reviewer_uw77 · 2024-03-11

**Summary Of Contributions:**

This paper mainly focuses on Active Learning. The authors mainly address three issues in the BADGE method: 1) the High computational complexity of gradient embedding in dense prediction tasks, 2) restrictions on the loss function of prediction labels, and 3) the inability to handle the regression problem well. A Hybrid method is proposed in the paper, which considers both uncertainty and diversity in sample selection, known as UWE. UWE mainly consists of two steps: 1) computing a novel uncertainty-weighted embedding and 2) applying distance-based sampling for sample selection. UWE has achieved state-of-the-art in three main visual tasks: image classification, semantic segmentation, and object detection. UWE is simple, not limited by loss functions or tasks, and has superior performance.

**Audience:**

Yes

**Broader Impact Concerns:**

None.

**Claims And Evidence:**

Yes

**Requested Changes:**

Please see the weaknesses above.

**Strengths And Weaknesses:**

Strengths:
1. Originality. UWE is very innovative. The author uses the chain rule to decompose gradient embedding into two terms: the first term depends on the form of the loss function and activation function; the second term is the feature representation extracted from the penultimate layer and is independent of the loss function. Therefore, the authors define a function u (x) to measure the uncertainty of the current model on sample x, and define a feature extractor f (x) by obtaining the intermediate output of the network, which is independent of loss functions. This decomposition method is novel and practical.
2. Quality. The paper conducts comprehensive experiments. Firstly, the authors conduct a correlation experiment between uwe (x), ge (x), and uncertainty, demonstrating a strong positive correlation between uwe (x) and uncertainty. Secondly, the author conduct experiments on three common visual tasks, showing the superior performance of UWE compared to other methods. Finally, the authors conduct ablation experiments to demonstrate the effectiveness of both parts of the UWE, and quantitatively compare the computational complexity with BADGE.
3. Clarity. The writing is clear. The authors start with the decomposition of gradient embedding and decompose it into a feature representation related to the loss function and activation function, as well as a feature representation unrelated to loss. Based on this decomposition, uncertainty-weighted embedding is proposed, which consists of two parts: u (x), which is positively correlated with uncertainty, and feature extractor f (x). The motivation of ideas is very intuitive, and the writing is clear.
4. Significance. UWE has good significance for the Active Learning community. This paper proposes a method with fewer restrictions, more flexibility, and better performance compared to BADGE, which can be migrated to many other tasks.

Weaknesses:
1. It is difficult to distinguish these methods in (a) and (b) of Figures 3, 4, 5, and 6. Especially, the lines corresponding to the better-performing methods almost overlap. It would be better if there were other high-contrast display methods available.
2. For image classification, it would be more convincing to add the experiment on ImageNet.
3. The authors simply summarize the results of different tasks, but a deeper analysis is missing but required, such as why the proposed method works and why it works better than existing methods.
4. Is it possible to give some visualizations, such as the detailed detection results, to better demonstrate the effectiveness of the proposed method?

---

> ### Author Response · Authors · 2024-04-14
>
> We would like to thank the reviewer for appreciating the originality, quality, clarity and significance of our work. Our responses to the weaknesses and requested changes are provided as below.
>
> **Presentation of benchmarking results**
>
> We plot the accuracy as a function of annotation budgets in Fig.3,4,5 and 6, which is a common practice in the literature. As we benchmark with many state-of-the-art techniques, the lines of those better performing methods almost overlap. To more clearly showcase the strength of method, we provide the pair-wise penalty matrices (PPM) proposed by Ji et al. (2023) that summarize the comparison results as a single numerical value in panels (c) of Figs.3,4 and 5. We also provide the raw data for Figs 3, 4, and 5 in A.7 of the revised paper for reference.
>
> **Results on ImageNet for image classification**
>
> Due to time and resource constraints, we report the results on mini-Imagenet instead in A.5 of the revised paper. We observe that UWE performs slightly better than BADGE, and outperforms others significantly. These results demonstrate the robustness of our method.
>
> **Deeper analysis on why the proposed method works**
>
> We conduct two more analyses to shed insight into why UWE works well. First, we compute the correlation between real gradient (RG) and UWE/pseudo gradient (PG, gradient computed with pseudo label as done in BADGE) in Section 5.4 of the revised paper. We observe that UWE obtains similar, or sometimes higher correlation coefficients than PG, suggesting that UWE performs comparably to PG regarding to approximating RG. Second, to validate that our method is indeed able to select a set of both uncertain and diverse samples, we compare the uncertainty and diversity of samples selected by different methods in A.4 of the revised paper. We observe that UWE ranks the second best  with regards to uncertainty, just behind Uncertainty sampling, which selects samples of top-k uncertainty. However, Uncertainty sampling does not consider diversity, thus ranking bottom regarding to diversity for image classification. Our method, being a hybrid method, is still able to achieve good diversity for image classification and  ranks first for object detection.
>
> **Visualization of detection results**
>
> We provide detailed detection results of models trained with samples selected by different active learning methods in A.6 of the revised paper.  We observe that our method is able to obtain detection results of higher quality, with fewer false positives and false negatives.

---

### Review · Reviewer_PkSp · 2024-03-14

**Summary Of Contributions:**

This paper explores the active learning problem, focusing on the iterative selection of informative unlabeled data points for labeling. The current state-of-the-art (SOTA) algorithm, BADGE, employs gradient embedding to identify uncertain samples, primarily tailored for classification tasks relying on cross-entropy (CE) loss. However, this paper extends the concept of gradient embedding to accommodate various loss functions. To achieve this, the authors decompose the gradient vector into an uncertainty measure and the feature extractor, inspired by the original BADGE embedding, resulting in what is termed uncertainty-weighted embedding (UWE). Subsequently, leveraging UWEs, the authors propose a distance-based sampling approach.

**Audience:**

Yes

**Claims And Evidence:**

No

**Requested Changes:**

Despite achieving state-of-the-art performance across various vision-related tasks with the introduction of UWE, the novelty of this paper remains somewhat limited. The authors should provide a detailed exploration of potential areas for increased novelty or the development of novel methodologies to enhance the paper's contribution.

**Strengths And Weaknesses:**

Strengths:

1. The simplicity and effectiveness of uncertainty-weighted embedding (UWE) extend beyond image classification to encompass image-level and region-level active learning tasks such as semantic segmentation and object detection.

2. In comparison to BADGE, uncertainty-weighted embedding offers advantages in terms of memory and computational efficiency.

Weaknesses:

1. The novelty of the paper appears limited. While the authors introduce uncertainty-weighted embedding (UWE), the overall methodology incorporates elements from existing approaches.

2. The algorithm relies heavily on heuristics, lacking thorough justification for its components. A more comprehensive discussion, including theoretical support, could bolster the paper's strength.

---

> ### Author Response · Authors · 2024-04-14
>
> We would like to thank the reviewer for appreciating the simplicity, effectiveness and efficiency of our method. Our responses to the weaknesses and requested changes are provided as below.
>
> **Novelty is limited**
>
> We believe incorporating elements from existing approaches does not necessarily translate into limited novelty, as these elements can be incorporated in an interesting way and new insights can be derived from the combination of existing approaches. Our method offers a novel perspective of viewing BADGE as method of "performing distance-based sampling in an uncertainty-aware feature space'', and provides an insightful two-term decomposition of the gradient embedding to motivate the design of a more flexible uncertainty-aware feature space. None of these ideas have been explored in previous works.
>
> **Justification of components**
>
> Our method consists of two major components: uncertainty-weighted embedding (UWE) extraction and distance-based sampling by K-Center-Greedy (KCG). The design of UWE is inspired by the two-term decomposition of the gradient embedding, and we show in Section 5.4 of the revised paper that it performs comparably to BADGE's gradient (gradient computed with pseudo label as ground truth label is not available for unlabelled data) regarding to approximating the true gradient. The choice of KCG over K-Means++ seeding algorithm is based on extensive comparison experiments reported in A.1, where we find that KCG works better for both UWE and gradient embedding on all the three tasks investigated in this paper.
>
> **Lacking theoretical support**
>
> We agree with the reviewer that providing comprehensive mathematical support can enhance the contribution of the paper. However, we believe the empirical evidence we present is still of interest to the AL community, especially considering the need to deploy AL in large scale real-world applications where flexibility and efficiency can be major concerns.

---

### Review · Reviewer_dMNC · 2024-03-29

**Summary Of Contributions:**

The paper is proposing a method for active learning based on the embedding values weighted by the uncertainty associated with them. Using such a modeling, which is related to the gradient-based approach proposed in BADGE, an iterative selection method is used to select the samples to label in the active learning process. This selection is based on the selection of the sample the furthest away from the other samples in the uncertainty-weighted embedding, in order to maximize diversity. That makes a specific uncertainty-diversity tradeoff in a given active learning routine, with results reported for image classification, semantic segmentation, and object detection.

**Audience:**

Yes

**Broader Impact Concerns:**

There are no ethical implications in this paper that would require a Broader Impact Statement.

**Claims And Evidence:**

No

**Requested Changes:**

First, I found the overall proposal not super clear nor well supported, explanations are relatively loose over the proposal and I am not convinced by the reasoning. For example, Sec. 3.2 presents the proposed UWE method, showing that we can use $\frac{\partial \ell}{\partial z}$ as an uncertainty measure. Then, it states “This motivates us to relax the constraint of computing the embedding as the derivative of loss function, and to generate an embedding by directly weighting the feature by uncertainty.” And then, the uncertainty function $u(x)$ is used as a replacement for $\frac{\partial \ell}{\partial z}$. So far, that makes sense, but it appears simply algebraic manipulations. What I got from that in the later experiment presentation is that from these manipulations, arbitrary uncertainty functions $u(x)$ are defined for each case tested, in replacement of the gradient $\frac{\partial \ell}{\partial z}$. I mean, why doing so, and what’s the issue in using the gradient, which is more exact? All this appears quite ad hoc and not very well supported, moreover given that we already have the analytical version available easily, as the gradients.

Also, the other part of the proposal is to make a greedy sequential selection of the instances to label from the current batch – see Algo. 1. This is a simple heuristic that may work reasonably well, but still may get stuck with a suboptimal set of instances given the myopic sequential selection done, where interactions between the instances selected is not taken into account globally, but rather after each iteration. Compared to this, the k-means++ algorithm used in BADGE appears to me better suited and less prone to provide suboptimal selection.

In Sec. 5.1, arguments on the computational advantages of UWE vs BADGE are made, which seems to be the main point supporting the approach. However, I am not sure about the claims made there, as everything seems to come from the fact that the uncertainty function approximates the gradient in some way, without being furthermore justified when they are presented (in section 3.4). So, basically, UWE is more efficient, computationally speaking, than BADGE as it relies on some heuristics that have been proposed with little justifications, whose heuristics are more efficient than computing the full gradients at hand. This is somehow expected, but the real question is whether the heuristics are **good** approximation of the real gradients. It might be so if we look at the results, but this is not studied in a direct and systematic way in the paper.

Comments on presentation:
- The bibliographic references are not well formatted in the document, with a lot of repetition of author names, having those written in the text and repeated from the citation. That’s almost everywhere in the paper, I would have expected the authors to be much more careful on that, a proofreading of the paper before the submission would have allowed the authors to catch that. Moreover, depending on the context, references should put the names and date in parentheses (e.g., (Smith et al., 2023)) and not names inline in the text (e.g., Smith et al. (2023)).
- Captions are not providing sufficient description of the figures and table to allow proper interpretation of those, in particular for figures 3 to 5, where panels (c) are not clear to me even after looking at the text.

**Strengths And Weaknesses:**

Strengths:
- Relevant topic (active learning), with the proposal of a complete method addressing it.
- Results on three distinct and relevant computer vision problems (image classification, semantic segmentation, object detection), with comparison with many techniques.


Weaknesses:
- The paper is not always easy to follow and read, the overall writing quality is fairly low, and the ideas are not presented very clearly.
- The working principles are relatively ad hoc, based on some observations on equations that are simplified with no clear justifications.
- It is argued that one advantage of the approach is the computational speedup in training such a model. These gains are, however, relatively modest (linear) in my opinion and come from simplifying the gradients by heuristics that are not themselves well justified and evaluated. For classification for example, the simplification mostly comes from the fact that BADGE will multiply the output by the number of classes in the dataset. This justification appears weak to me, and the gains are factors of the number of classes, not in order of magnitudes.
- The results are not easy to interpret and the advantage of the proposed approach over the others is not very clear. I am in fact doubtful of the gains of the proposed UWE over the BADGE approach, as UWE appears to be an approximation version of BADGE, using custom uncertainty measure vs real gradients, and making a sequential greedy selection of samples to label vs using k-means++.

---

> ### Author Response · Authors · 2024-04-14
>
> We would like to thank the reviewer for acknowledging the strengths of our method, including  being a relevant and complete method for AL, providing results on three distinctive and relevant computer vision problems with comparison with many techniques. Our responses to the weaknesses and requested changes are provided as below.
>
> **Why use UWE instead of gradient**
>
> One issue with using the gradient is that it relies on pseudo labels for the unlabeled data so it is also an approximation of the true gradient. We compute the correlation between real gradient (RG) and UWE/pseudo gradient (PG, gradient computed with pseudo label as done in BADGE) in Section 5.4 of the revised paper. We observe that UWE obtains similar, or sometimes higher correlation coefficients than PG, suggesting that UWE performs comparably to PG regarding to approximating RG.
>
> Another issue is that BADGE was only tested on image classification tasks. When we apply BADGE for object detection, which involves both classification and regression tasks, it underperforms our method significantly. We also observe that using the gradient embedding from the regression branch or region proposal network harms the performance (detailed experimental results are provided in A.3 of the revised paper).
>
> These observations suggest that BADGE’s gradient embedding is not guaranteed to work well in practice, and our method with task-aware uncertainty and feature design can still outperform BADGE.
>
> **K-Means++ (KM++) vs. K-Center-Greedy (KCG)**
>
> We would like to clarify that the K-Means++ algorithm used in BADGE refers to the K-Means++ seeding algorithm (refer to Algorithm 2 of the BADGE paper for more details), i.e., the algorithm used to select the initial centers for the following K-Means clustering, not the full clustering algorithm. K-Means++ seeding algorithm is also a sequential greedy selection algorithm like KCG. The only difference between KM++ seeding algorithm and KCG is that the former selects samples probabilistically with probability proportional to the squared distance to the selected samples in each iteration, while the latter deterministically selects the sample with the largest distance. We conduct extensive experiments to compare KM++ seeding algorithm and KCG in A.1 of the revised paper (Section 7.1 of the original paper), and find that KCG works better for both UWE and gradient embedding on all the three tasks investigated in this paper. We thus adopt KCG in our method.
>
> **Computation efficiency of UWE**
>
> We would like to highlight that UWE can be faster by factors larger than the number of classes depending on the specific network architecture. As discussed in Section 5.1, the time complexity of KCG and KM++ seeding algorithm is $O(n_b\cdot n_u \cdot n_{dim})$, where $n_b$ is the number of selected samples, $n_u$ is the total number of unlabelled samples, and $n_{dim}$ is the dimension of the sample. For image classification and semantic segmentation, $n_{dim}$ for UWE is the same as feature dimension $f_{dim}$, while the dimension of the gradient embedding used in BADGE is $n_c\cdot f_{dim}$, where $n_c$ is the number of classes. As such, our method is more efficient than BADGE by a factor of $n_c$. For object detection, $n_{dim}$ for UWE is $n_c\cdot f_{dim}$ due to class-wise aggregation. The dimension of gradient embedding, however, varies depending on detector architectures and is typically much higher as it not only depends on the number of classes, but also other network parameters. For example, the dimension of gradient embedding for RetinaNet is $k^2\cdot n_a\cdot n_c\cdot f_{dim}$, where $n_a=9$ is the number of anchors and $k=3$ is the filter size for the prediction head. As such, our method is more efficient than BADGE by a factor of $n_a*k^2=81$, though $n_c$ is 20 on VOC0712.
>
> **Presentation issues**
>
> We have fixed the citation format problem in the revised paper. As for the panels (c) in Figs.3-5, they are pair-wise penalty matrices (PPM) proposed by Ji et al. (2023). We have modified the first paragraph of Section 4 in the revised draft to provide more computation details of PPM.

---

### Decision · Action_Editor_zrLc · 2024-05-11

**Recommendation:** Accept with minor revision

**Comment:**

This paper proposes an uncertainty-based strategy for active learning. The final recommendations from the three reviewers are at odds: two weak accepts and one weak reject. By inspecting all reviews and the revised paper, the AE thought that this submission can be accepted contingent on minor revisions. The merits of the method lie in the high efficiency, the extensibility to different tasks, the capability to handle object detection and semantic segmentation tasks with improved performances, and the insights into the decomposition of the gradient-based criterion. However, the paper shall undergo a minor revision, better addressing the comments from Reviewer #dMNC. In particular, the rationale behind the decomposition of gradient-based criterion, i.e. into an uncertainty measure and a feature extractor, shall be elaborated further; Some empirical evidence shall be provided that supports each part of the decomposition.

**Audience:**

TMLR's audience will be interested in this submission, especially those doing research in active learning and weakly-supervised learning, and those applying these techniques to real-world applications such as object detection.

**Claims And Evidence:**

The claims made in the submission are well supported by convincing empirical evidence. While the theoretical perspective is relatively lacking, but since this is an empirical study, it is acceptable.